# CDK9-dependent RNA polymerase II pausing controls transcription initiation

Saskia Gressel[1†], Björn Schwalb[1†*], Tim Michael Decker[2], Weihua Qin[3], Heinrich Leonhardt[3], Dirk Eick[2*], Patrick Cramer[1*]

[1]Department of Molecular Biology, Max-Planck-Institute for Biophysical Chemistry, Göttingen, Germany; [2]Department of Molecular Epigenetics, Helmholtz Center Munich, Center of Integrated Protein Science, Munich, Germany; [3]Department of Biology II, Ludwig-Maximilians-Universität München, Center of Integrated Protein Science, Martinsried, Germany

*For correspondence:
bjoern.schwalb@mpibpc.mpg.de (BS);
eick@helmholtz-muenchen.de (DE);
patrick.cramer@mpibpc.mpg.de (PC)

†These authors contributed equally to this work

Competing interests: The authors declare that no competing interests exist.

**Abstract** Gene transcription can be activated by decreasing the duration of RNA polymerase II pausing in the promoter-proximal region, but how this is achieved remains unclear. Here we use a 'multi-omics' approach to demonstrate that the duration of polymerase pausing generally limits the productive frequency of transcription initiation in human cells ('pause-initiation limit'). We further engineer a human cell line to allow for specific and rapid inhibition of the P-TEFb kinase CDK9, which is implicated in polymerase pause release. CDK9 activity decreases the pause duration but also increases the productive initiation frequency. This shows that CDK9 stimulates release of paused polymerase and activates transcription by increasing the number of transcribing polymerases and thus the amount of mRNA synthesized per time. CDK9 activity is also associated with long-range chromatin interactions, suggesting that enhancers can influence the pause-initiation limit to regulate transcription.
DOI: https://doi.org/10.7554/eLife.29736.001

## Introduction

Transcription in metazoan cells is often regulated at the level of promoter-proximal pausing (*Core et al., 2008*; *Day et al., 2016*; *Henriques et al., 2013*; *Nechaev et al., 2010*; *Rougvie and Lis, 1988*; *Strobl and Eick, 1992*), which can be detected by measuring the occupancy with paused Pol II by ChIP-seq (*Johnson et al., 2007*), GRO-seq (*Core et al., 2008*), (m)NET-seq (*Mayer et al., 2015*; *Nojima et al., 2015*), or PRO-seq (*Kwak et al., 2013*). Genes with paused Pol II are conserved across mammalian cell types and states (*Day et al., 2016*). The mechanisms underlying how Pol II pausing can regulate RNA transcript synthesis remain unclear.

Transcription of a human protein-coding gene of average length takes at least half an hour to be completed. The duration of pausing however lies in the range of minutes (*Jonkers et al., 2014*) and does not considerably change the overall time it takes to complete a transcript. Thus, how can changes in the pause duration lead to synthesis of a different number of RNA transcripts per time? It has been suggested that a decreased pause duration goes along with a higher initiation frequency, because occupancy peaks for promoter-proximal Pol II can increase upon gene activation (*Boehm et al., 2003*) or can remain high even when pausing is impaired (*Henriques et al., 2013*).

The height of Pol II occupancy peaks however cannot directly inform on initiation frequency or pause duration because it depends not only on the number of polymerases that pass the pause site but also on their residence time (*Ehrensberger et al., 2013*). A kinetic model of transcription predicted that pause duration delimits the initiation frequency and suggested that paused Pol II sterically interferes with initiation (*Ehrensberger et al., 2013*). Indeed, modeling reveals that a paused polymerase positioned up to around 50 bp downstream of the TSS could sterically interfere with

**eLife digest** Genes can contain the coded instructions to make proteins. These instructions must first be copied, or transcribed, into an intermediate molecule called a messenger RNA by an enzyme known as RNA polymerase II. Shortly after it begins, this enzyme – which is called Pol II for short – pauses, and it only starts again after it recruits other proteins, including one called CDK9.

The number of RNA copies made of a gene depends upon how many Pol II enzymes begin transcription. Pol II pausing also has an effect – if the enzymes pause for longer, less messenger RNA is transcribed. But why does this happen? One hypothesis is that paused Pol II enzymes interfere with other Pol II enzymes initiating transcription. Yet, until recently it was not possible to measure if this actually happens in living cells.

Now, Gressel, Schwalb et al. used a new biochemical method together with a compound that blocks CDK9 to measure pausing and transcription initiation for active genes in living human cells. The CDK9 inhibitor was used to make Pol II enzymes pause for longer than normal. Gressel, Schwalb et al. found that different genes responded differently to CDK9 inhibition, meaning that some remained paused for longer than others. The number of Pol II enzymes that initiated transcription was calculated by measuring how many RNA copies had been made locally at that the site of transcription. These experiments showed that blocking the release of paused Pol II strongly reduced the number of RNA copies made.

Gressel, Schwalb et al. conclude that Pol II pausing can control initiation of transcription. Cells may use Pol II pausing to adjust how many copies of an RNA are made, helping to ensure that different cell types make the appropriate number of RNA copies from a gene. Many diseases are associated with gene transcription being incorrectly regulated. This and future studies will help scientists to better understand how Pol II pausing contributes to the control of transcription in both normal and diseased cells.

DOI: https://doi.org/10.7554/eLife.29736.002

formation of the Pol II initiation complex (*Figure 1—figure supplement 1*). Even if a paused polymerase is located further downstream, it may still interfere with initiation if one or more additional elongating polymerases line up behind it.

The critical relationship between pausing and initiation could thus far not be tested experimentally, as no methods were available to measure initiation frequencies. A recently developed method, transient transcriptome sequencing (TT-seq) (*Schwalb et al., 2016*), now allows to unveil the flow of polymerases as it measures local RNA synthesis rates genome-wide at nucleotide resolution.

Here we investigate whether changes in pause duration alter initiation frequency in living cells. We specifically inhibit the kinase CDK9, which facilitates Pol II pause release (*Laitem et al., 2015*; *Marshall and Price, 1992*; *Peterlin and Price, 2006*), and monitor RNA synthesis and initiation frequencies by TT-seq. A combination of TT-seq data with mNET-seq data allows us to derive pause durations for active genes. We conclude that the duration of pausing can control transcription initiation at human genes, and derived determinants for CDK9-dependent pause release and initiation activation.

## Results

### CRISPR-Cas9-engineered mutation allows for specific CDK9 inhibition

To specifically inhibit CDK9, we used a chemical biology approach (*Lopez et al., 2014*) that circumvents off-target effects of standard CDK9 inhibitors (*Morales and Giordano, 2016*). We introduced a CDK9 analog sensitive mutation (CDK9[as]) into human Raji B cells by CRISPR-Cas9 (Materials and methods, *Figure 1—figure supplement 2A–B*). This allows for rapid and highly specific CDK9 inhibition with the adenine analog 1-NA-PP1 (*Lopez et al., 2014*), which does not have any effect on wild type cells (*Figure 1—figure supplement 2C*). CDK9 protein levels were unchanged in CDK9[as] mutant cells compared to wild type cells (*Figure 1—figure supplement 2D*). After 72 hr of incubation with 1-NA-PP1, growth of CDK9[as] cells ceased, whereas wild type cells grew normally (*Figure 1—figure supplement 2E*).

## TT-seq monitors immediate response to CDK9 inhibition

We treated CDK9[as] cells with 5 μM of 1-NA-PP1 for 10 min and monitored changes in RNA synthesis by TT-seq (*Schwalb et al., 2016*), using a RNA labeling time of 5 min (*Figure 1A*). TT-seq data were highly reproducible (Spearman correlation coefficient 1) and monitored transcription activity before and after CDK9 inhibition (*Figure 1B*). CDK9 inhibition resulted in reduced TT-seq signals at the beginning of genes, indicating that less Pol II was released into gene bodies (*Figure 1B*, *Figure 2— figure supplement 1A–B*). This gave rise to a 'response window' revealing the distance traveled by Pol II during 10 min inhibitor treatment (*Figure 1C*). Downstream of the response window, the TT-seq signal was largely unchanged, indicating continued RNA synthesis from Pol II elongation complexes that had been released before CDK9 inhibition.

To determine the relative response of genes to CDK9 inhibition, we calculated response ratios for those transcribed units (TUs, Materials and methods) that synthesized RNA, harbored a single TSS, and exceeded 10 kbp in length (2,538 TUs). The response ratio of TUs varied between 0% to 100% (fully responding TUs) with a median of 58% (*Figure 1C–E*). A remaining TT-seq signal in the response window likely reflects the proportion of polymerases that move to productive elongation without CDK9 kinase activity, but we cannot exclude that it stems from incomplete CDK9 inhibition. However, based on the assumption that the inhibitor is evenly distributed across cells and within, the portion of CDK9 that has not been fully inhibited must be very low.

## Pol II elongation velocity is gene-specific

The width of the response window differs between TUs (*Figure 1D*) and informs on Pol II elongation velocity (Materials and methods). The average width of the response window was 23 kbp, and thus the average elongation velocity was 2.3 kbp/min (*Figure 2A–B*), which agrees with previous estimates (*Fuchs et al., 2014*; *Jonkers et al., 2014*; *Saponaro et al., 2014*; *Veloso et al., 2014*). Gene-specific elongation velocities (*Figure 2C*, *Figure 2—figure supplement 1A–B*) were significantly higher in TUs with longer first introns (*Figure 2D*, Wilcoxon rank sum test, p-value$<1.916 \cdot 10^{-11}$), consistent with faster transcription of introns (*Jonkers et al., 2014*). Elongation velocity correlated positively with nucleosome density, and negatively with the stability of the DNA-RNA hybrid, CpG density and topoisomerase occupancy (*Figure 2—figure supplement 1C*).

## Promoter-proximal pausing occurs at sequences that give rise to weak DNA-RNA hybrids

To study the kinetics of CDK9-dependent Pol II pause release, we generated mNET-seq data that map the RNA 3'-end of engaged Pol II and extracted the position of paused polymerases (Materials and methods). mNET-seq data were highly reproducible (Spearman correlation coefficient 0.93). Of the above TUs, 2135 (84 %) showed mNET-seq signal peaks above background (Materials and methods). The called pause sites were distributed around a maximum located ~84 bp downstream of the TSS (*Figure 3A*, *Figure 3—figure supplement 1A*). At these sites we detected an enrichment for G/C-C/G dinucleotides (*Figure 3—figure supplement 1B*) with a strongly conserved cytosine at the RNA 3'-end (*Figure 3B*). We also observed a minimum of the predicted melting temperature of the DNA-RNA hybrid (Materials and methods) immediately downstream of the pause site (*Figure 3C*). A weak DNA-RNA hybrid in the active center of Pol II is known to destabilize the elongation complex (*Kireeva et al., 2000*), and could be a major determinant for establishing the paused state.

## Multi-omics analysis provides pause duration *d* and initiation frequency *I*

To quantify pausing, we defined the pause duration *d* as the time a polymerase needs to pass through a 200 bp 'pause window' located ±100 bp around the pause site. The pause duration *d* can now be derived from a combination of mNET-seq and TT-seq data. In particular, the mNET-seq signal corresponds to the number of polymerases in the pause window, which is determined by *d* and by the initiation frequency *I* (*Figure 4A*) (*Ehrensberger et al., 2013*). Thus, *d* is proportional to the ratio of the mNET-seq signal over *I*. To calculate *I* we integrated TT-seq signals over exons, excluding the first exon (Materials and methods). This provides the 'productive initiation frequency', that is the number of polymerases that initiate and successfully exit from the pause window. We use the term 'productive' because we do not know whether there is a small fraction of polymerases

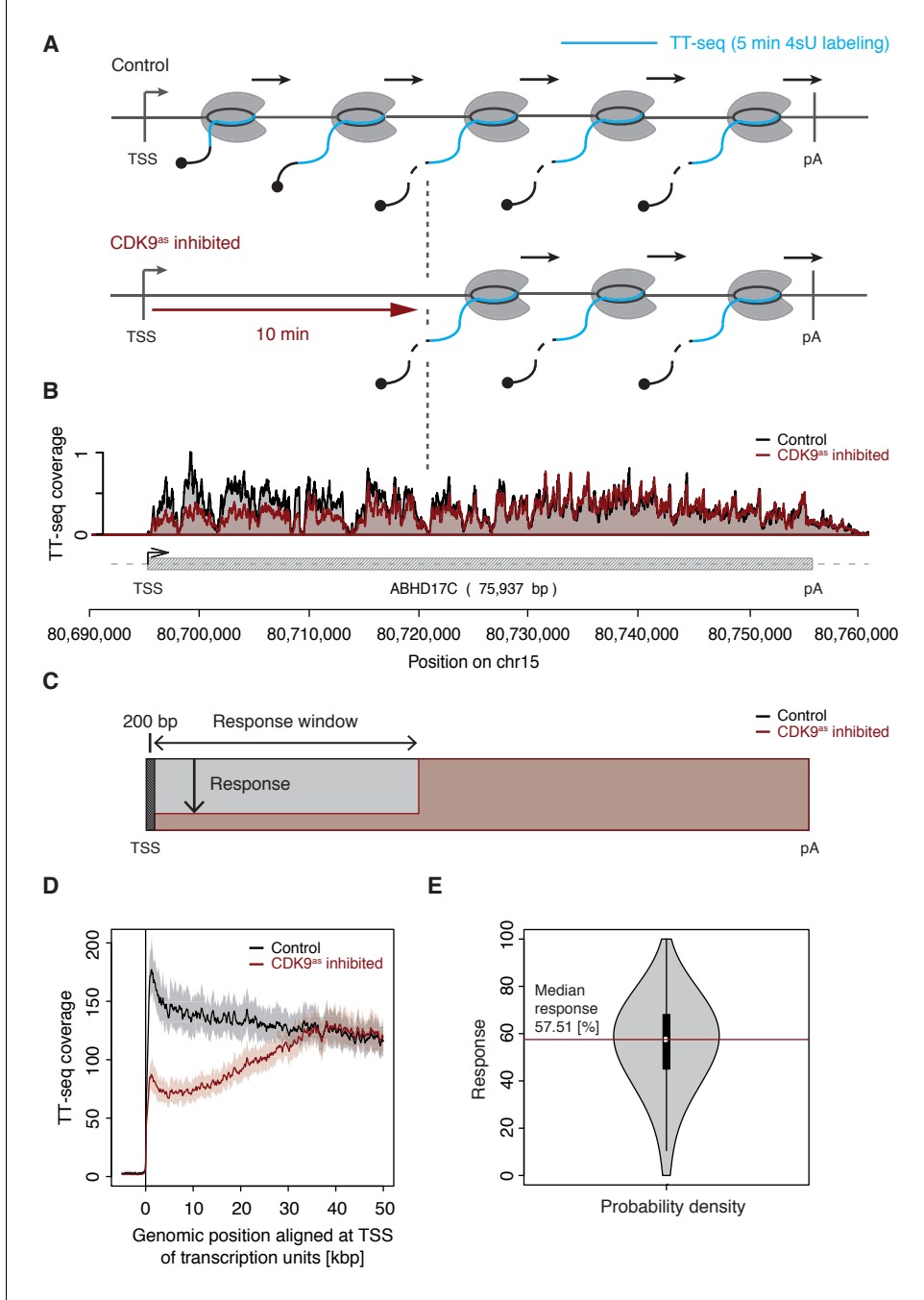

**Figure 1.** CDK9 inhibition decreases RNA synthesis in the 5'-region of genes. (**A**) Experimental design. TT-seq was carried out with CDK9[as] cells after treatment with solvent DMSO (control) or 1-NA-PP1 (CDK9[as] inhibited). (**B**) TT-seq signal before (black) and after (red) CDK9 inhibition at the ABHD17C gene locus (75,937 [bp]) on chromosome 15. Two biological replicates were averaged. The grey box depicts the transcript body from the transcription start site (TSS, black arrow) to the polyA site (pA). (**C**) Schematic representation of changes in TT-seq signal showing the definition of the response window. Colors are as in (**B**). (**D**) Metagene analysis comparing the average TT-seq signal before and after CDK9 inhibition. The TT-seq coverage was averaged for 954 out of 2538 investigated TUs that exceed 50 [kbp] in length (Materials and methods). TUs were aligned with their TSS. Shaded areas around the average signal (solid lines) indicate confidential intervals (Materials and methods). (**E**) Violin plot showing the relative response to CDK9 inhibition for 2538 investigated TUs defined as 1 - (CDK9[as] inhibited/ Control) ·100 for a window from the TSS to 10 [kbp] downstream, excluding the first 200 [bp] (**C**). A red line indicates the median response (58%).

*Figure 1 continued on next page*

*Figure 1 continued*

DOI: https://doi.org/10.7554/eLife.29736.003

The following figure supplements are available for figure 1:

**Figure supplement 1.** Model of a paused polymerase positioned up to around 50 bp downstream of the TSS.
DOI: https://doi.org/10.7554/eLife.29736.004

**Figure supplement 2.** CRISPR-Cas9 directed engineering, cellular and biochemical characterization of CDK9[as] Raji B cell line.
DOI: https://doi.org/10.7554/eLife.29736.005

terminating within the pause window. Finally, to derive absolute values of *d*, we scaled the reciprocal

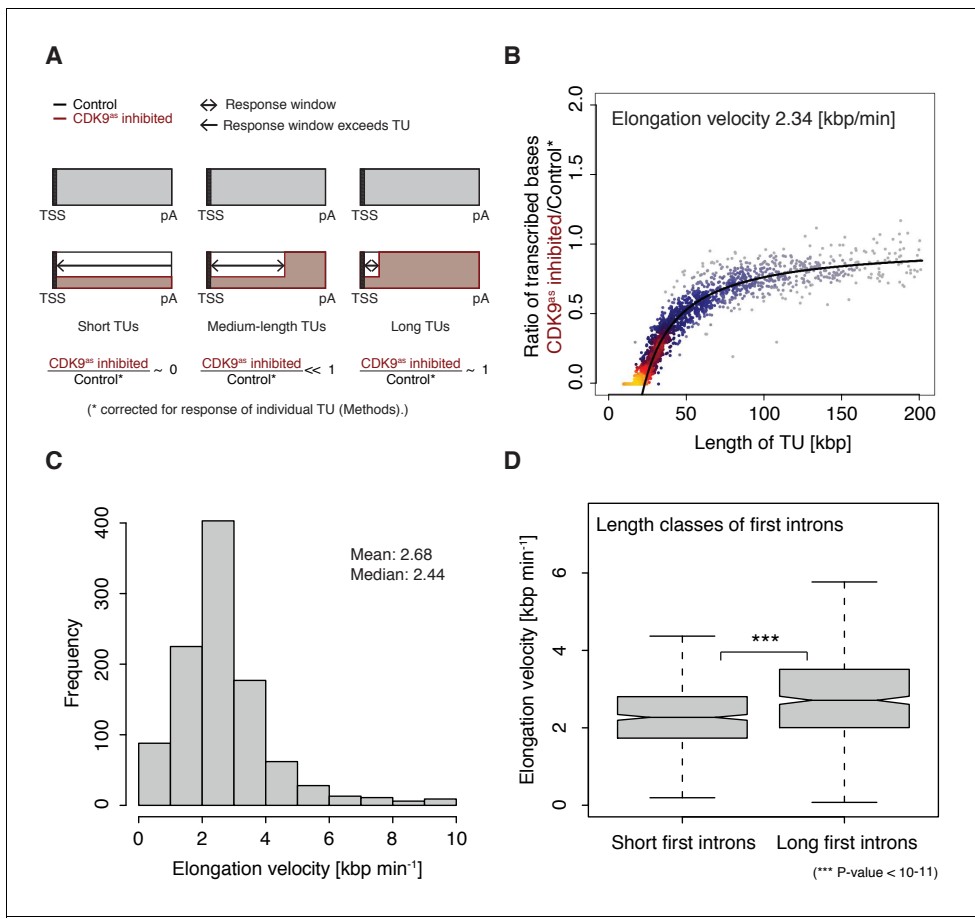

**Figure 2.** Pol II elongation velocity. (**A**) Schematic representation of observed response window of TT-seq signal with CDK9[as] inhibitor (red) or control (black) for TUs of three different length classes (short TUs < 25 [kbp], medium-length TUs 25–50 [kbp] and long TUs > 100 [kbp]). (**B**) Scatter plot of the ratio of transcribed bases (CDK9[as] inhibited/control) (Materials and methods) against the length of the TUs in nucleotides [kbp] revealed that the schematic representation in (**A**) holds true for 2443 investigated TUs (Materials and methods). Modeling of the observed relation allows estimation of a robust average elongation velocity of 2.3 [kbp/min] (solid black line, Materials and methods). (**C**) Distribution of gene-wise elongation velocity depicted as a histogram (mean 2.7 [kbp/min], median 2.4 [kbp/min]). (**D**) Distributions of elongation velocity [kbp/min] depicted for 513 TUs with short first intron (<50% quantile, left) and 514 TUs with long first intron (>50% quantile, right).

DOI: https://doi.org/10.7554/eLife.29736.006

The following figure supplement is available for figure 2:

**Figure supplement 1.** Example genome browser views of TT-seq signals in CDK9[as] cells with estimated response window and genomic features correlating with elongation velocity.
DOI: https://doi.org/10.7554/eLife.29736.007

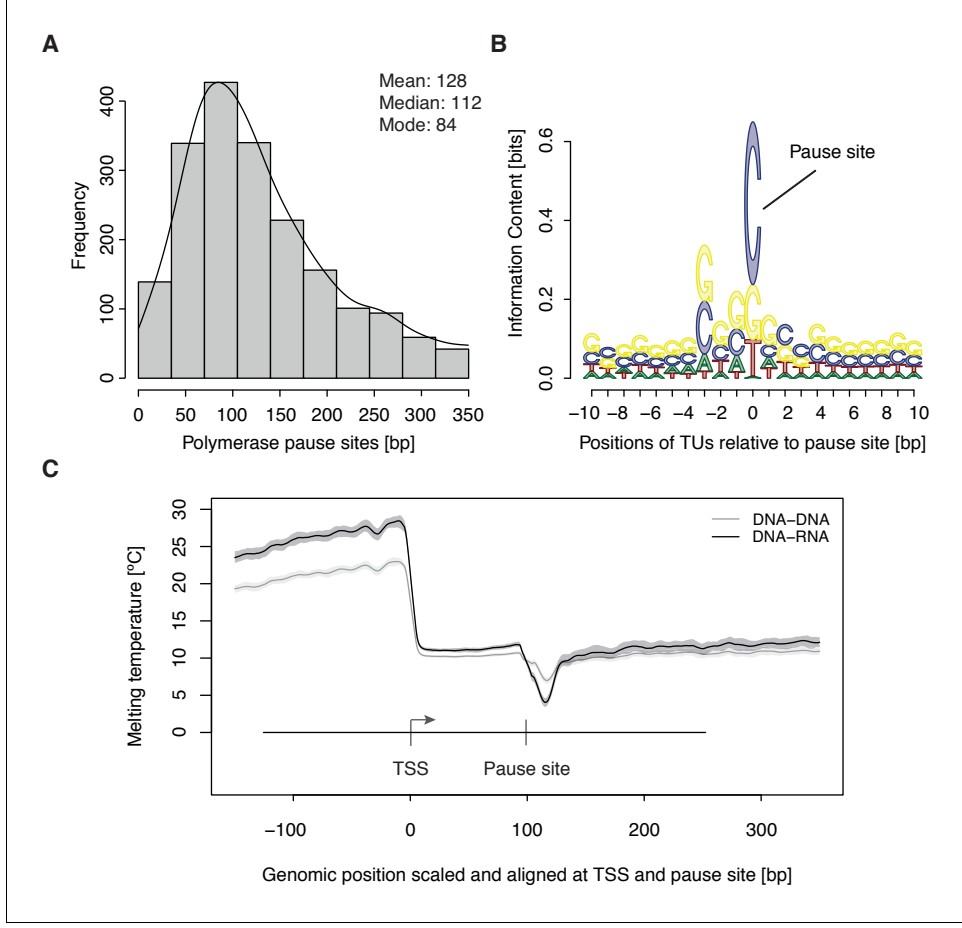

**Figure 3.** Distribution and sequence of promoter-proximal pause sites. (**A**) Distribution of pause site distance from the TSS for 2135 investigated TUs depicted as a histogram (mean 128 [bp], median 112 [bp], mode 84 [bp]). Two biological replicates were averaged. (**B**) Position weight matrix (PWM) logo representation of bases at positions –10 to +10 [bp] around the pause site (position 0). (**C**) Mean melting temperature of the DNA-RNA and DNA-DNA hybrid aligned at the TSS and the pause site (signal between the TSS and the pause site is scaled to common length of 100 [bp]). Shaded areas around the average signal (solid lines) indicate confidence intervals.
DOI: https://doi.org/10.7554/eLife.29736.008

The following figure supplement is available for figure 3:

**Figure supplement 1.** Features of underlying DNA sequence around promoter-proximal pause sites.
DOI: https://doi.org/10.7554/eLife.29736.009

of $d$ (the elongation velocity in the pause window) according to the elongation velocity obtained from CDK9 inhibition (Materials and methods).

We obtained a mean productive initiation frequency of 2.7 polymerases $cell^{-1}min^{-1}$, and pause durations in the range of minutes, with strong variations between TUs. The pause durations are generally consistent with reported half-lives of paused Pol II in mouse (*Jonkers et al., 2014*) and Drosophila cells (*Buckley et al., 2014*; *Henriques et al., 2013*) but slightly shorter. Pause durations were also consistent with kinetic modeling of TT-seq data alone. At TUs with long pause durations we observed less labeled RNA in the short region between the TSS and the pause site (*Figure 4—figure supplement 1*). This confirms that indeed initiation frequencies are altered. It also indicates that the fraction of Pol II enzymes that terminate within the pause window is low, in agreement with previous findings (*Henriques et al., 2013*). For strongly CDK9-responding TUs, we obtained a significantly longer pause duration (Wilcoxon rank sum test, p-value$<10^{-12}$) and lower initiation frequencies (*Figure 4B–C*).

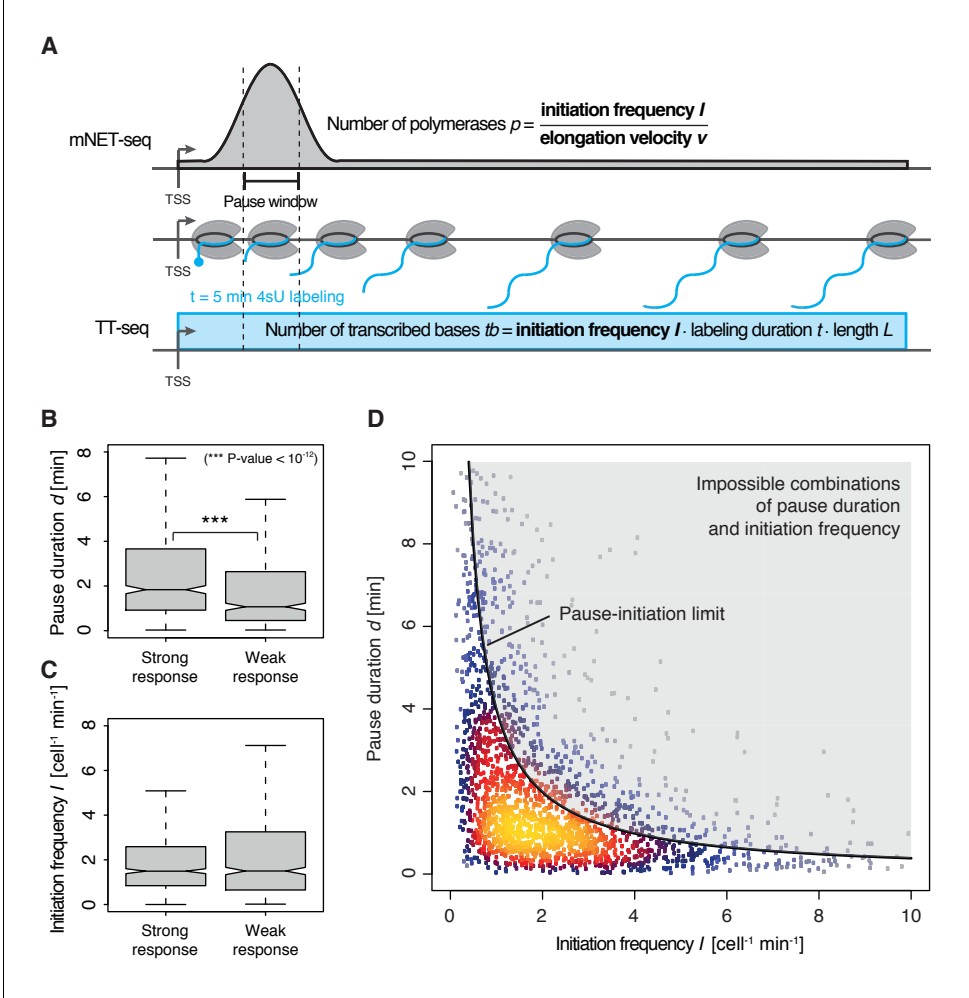

**Figure 4.** Pol II pausing generally limits transcription initiation ('pause-initiation limit'). (**A**) Schematic representation of polymerase flow in the promoter-proximal region. The mNET-seq signal (top) is the ratio of the initiation frequency $I$ over the elongation velocity $v$. The TT-seq signal (bottom) corresponds to initiation frequency $I$. Thus, $v$ can be derived from the ratio of the TT-seq over the mNET-seq signal, and the reciprocal of $v$ in the pause window corresponds to the pause duration $d$. (**B**) Distributions of gene-wise pause duration $d$ [min] for TUs with a CDK9 response ratio >75% quantile (574 TUs) and TUs with a response ratio <25% quantile (469 TUs). (**C**) Distributions of gene-wise initiation frequency $I$ [cell$^{-1}$min$^{-1}$] for TUs with a CDK9 response ratio >75% quantile (635 TUs) and TUs with a response ratio <25% quantile (635 TUs). (**D**) Scatter plot between the initiation frequency $I$ [cell$^{-1}$min$^{-1}$] and the pause duration $d$ [min] for 2135 common TUs with color-coded density estimation. The grey shaded area depicts impossible combinations of $I$ and $d$ according to published kinetic theory (**Ehrensberger et al., 2013**) and assuming that steric hindrance occurs below a distance of 50 [bp] between the active sites of the initiating Pol II and the paused Pol II.

DOI: https://doi.org/10.7554/eLife.29736.010

The following figure supplements are available for figure 4:

**Figure supplement 1.** A longer pause duration but not promoter-proximal termination of polymerase leads to shortage of labeled RNA in the region between TSS and pause site.

DOI: https://doi.org/10.7554/eLife.29736.011

**Figure supplement 2.** Verification of anti-correlation between initiation frequency $I$ and pause duration $d$ including 'pause-initiation limit'.

DOI: https://doi.org/10.7554/eLife.29736.012

## Human genes have a 'pause-initiation limit'

These results prompted us to ask whether the pause duration is generally related to the initiation frequency. We indeed found a robust anti-correlation between $I$ and $d$ in normally growing cells, and an upper boundary for combinations of $I$ and $d$ which we call 'pause-initiation limit'. (*Figure 4D*, *Figure 4—figure supplement 2A*). Thus, genes with shorter pausing show higher initiation frequencies and more RNA synthesis. This fundamental relationship can be verified by calculating the pause duration $d$ without the initiation frequency $I$, $\hat{d}$ (Materials and methods, *Figure 4—figure supplement 2B–C,E*). Repeated random shuffling of mNET-seq signal assignment to TUs abolishes the correlation between $\hat{d}$ and $I$ (*Figure 4—figure supplement 2D*). It also shows that the observation of impossible combinations of pause duration $d$ and initiation frequency $I$ (points above 'pause-initiation limit') are minimal (*Figure 4—figure supplement 2F*). In conclusion, independent mNET-seq and TT-seq data led to independent measures of pause duration and productive initiation frequency for each gene, which were then observed to be globally anti-correlated.

These findings now allowed us to test directly whether longer pause durations lead to lower initiation frequencies, by analyzing TT-seq data after CDK9 inhibition. CDK9 inhibition resulted in significantly reduced labeled RNA in the short region between the TSS and the pause site (Wilcoxon rank sum test, p-value$<10^{-16}$) (*Figure 5A–B*). Productive initiation frequencies were significantly downregulated after CDK9 inhibition (Wilcoxon rank sum test, p-value$<10^{-16}$) (*Figure 5C*). Because CDK9 specifically targets paused Pol II, and not initiating polymerase, these results show that pausing limits initiation, and not the other way around. Thus, human genes have a 'pause-initiation limit'.

To monitor the occupancy of engaged Pol II we generated mNET-seq data before and after CDK9 inhibition (Materials and methods). CDK9 inhibition resulted in increased mNET-seq signal at the beginning of genes and decreased signal in the gene body, indicating that less Pol II was

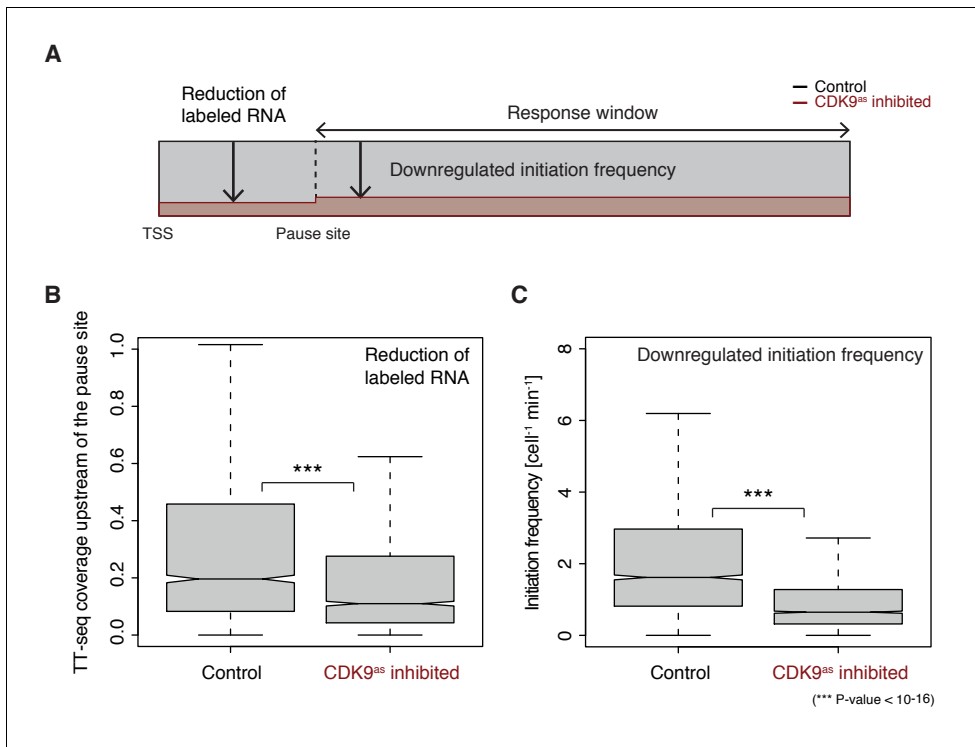

**Figure 5.** Increasing Pol II pause duration decreases the frequency of transcription initiation. (**A**) Schematic representation of observed decrease in TT-seq signal upon CDK9 inhibition, upstream and downstream of the pause site. (**B**) Distributions of gene-wise mean TT-seq signals in the region between the TSS and the pause site, before (control) and after CDK9 inhibition, normalized to the initiation frequency before CDK9 inhibition. (**C**) Distributions of gene-wise initiation frequencies before (control) and after CDK9 inhibition.

DOI: https://doi.org/10.7554/eLife.29736.013

released from the pause site (*Figure 6A*). Indeed, calculation of pause durations from mNET-seq and TT-seq data after CDK9 inhibition showed that Pol II resides significantly longer at the pause site after CDK9 inhibition (Wilcoxon rank sum test, p-value<$10^{-16}$) (*Figure 6B*). Taken together, CDK9 inhibition increases the pause duration and decreases the initiation frequency at human genes (*Figure 6C–D*).

## Determinants of promoter-proximal pausing

To investigate possible reasons for polymerase pausing and its consequences, we compared different properties of TUs with long and short pause durations. For the 5'-region of TUs with longer pause durations, the transcript adopts more RNA secondary structure in vivo and in silico (Wilcoxon rank sum test, p-value<$10^{-16}$) (*Figure 7A*, *Figure 7—figure supplement 1A*) (*Rouskin et al., 2014*). TUs with longer pause durations were also enriched for hyper-methylated CpG islands (*ENCODE Project Consortium, 2012*) upstream of the pause site (*Figure 7B*), consistent with a

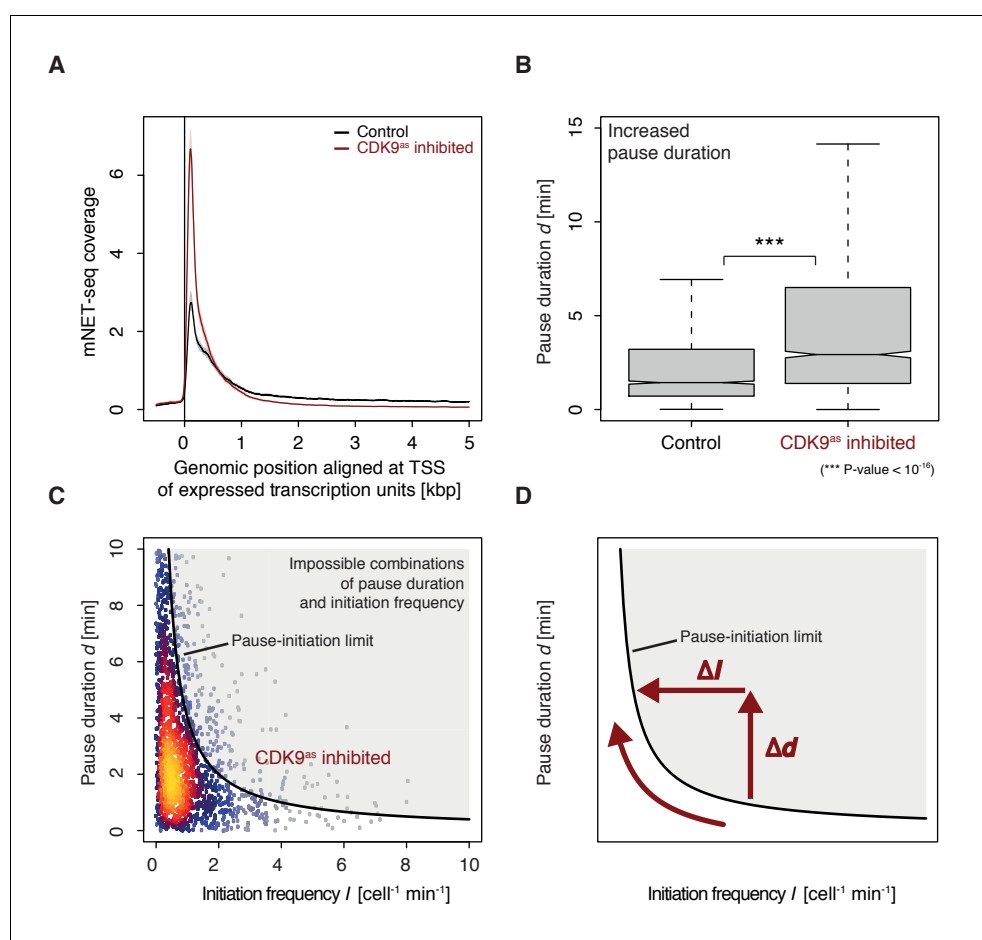

**Figure 6.** CDK9 inhibition leads to increased pause duration. (**A**) Metagene analysis comparing the average mNET-seq signal before and after CDK9 inhibition. Two biological replicates were averaged. The mNET-seq coverage was averaged for 2538 investigated TUs (Materials and methods). TUs were aligned with their TSS. Shaded areas around the average signal (solid lines) indicate confidentiality intervals (Materials and methods). (**B**) Distributions of gene-wise pause duration $d$ [min] before (control) and after CDK9 inhibition. (**C**) Scatter plot between the initiation frequency $I$ [cell$^{-1}$min$^{-1}$] and the pause duration $d$ [min] after CDK9 inhibition for 2135 common TUs with color-coded density estimation. The grey shaded area depicts impossible combinations of $I$ and $d$ (*Ehrensberger et al., 2013*) assuming that steric hindrance occurs below a distance of 50 [bp] between the active sites of the initiating Pol II and the paused Pol II. (**D**) Schematic of changes in pause duration ($\Delta d$) and initiation frequency ($\Delta I$) upon CDK9 inhibition. As a consequence, data points in panel (**D**) are moved to the left and upwards.

DOI: https://doi.org/10.7554/eLife.29736.014

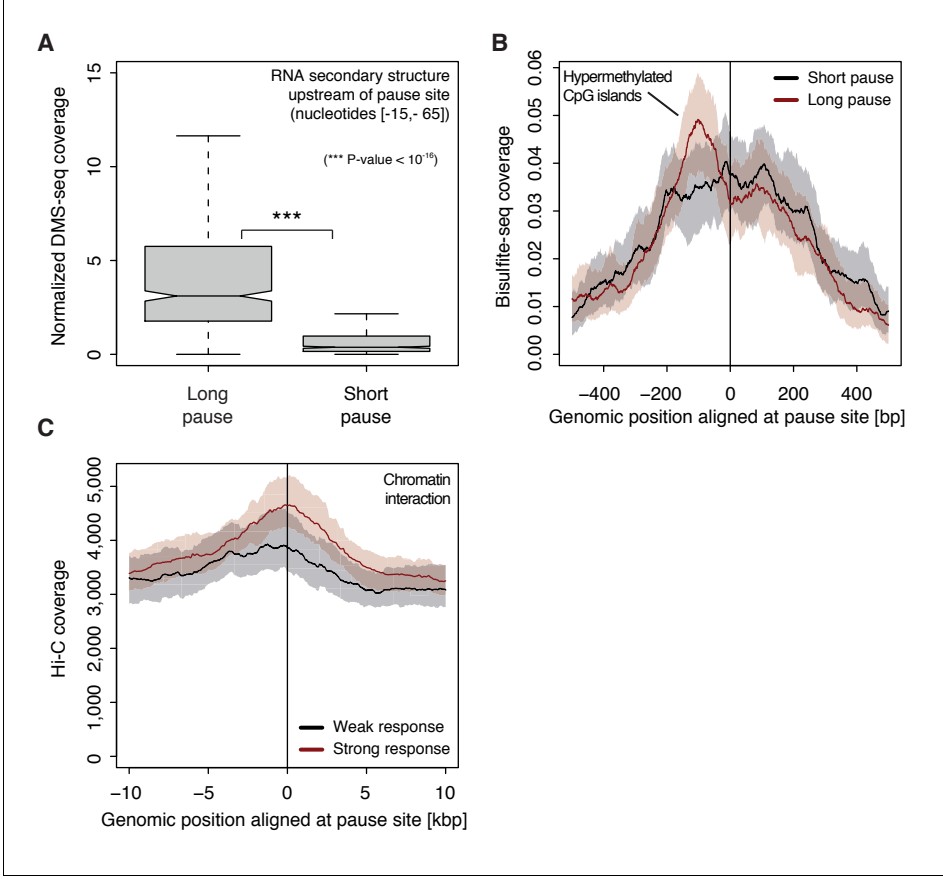

**Figure 7.** Determinants of CDK9-dependent promoter-proximal pausing. (**A**) Distribution of gene-wise mean in vivo DMS-seq signals (detecting RNA secondary structure) for a window between −65 and −15 [bp] upstream of the pause site for TUs with long pause durations (pause duration >75% quantile, 534 TUs) and with short pause durations (pause duration <25% quantile, 534 TUs) normalized to denatured DMS-seq coverage (Materials and methods). (**B**) Metagene analysis comparing the average Bisulfite-seq signal (detecting methylated DNA) for subsets as in (**A**) aligned at the pause site (red, long pause duration, and black, short pause duration). Shaded areas around the average signal (solid lines) indicate confidence intervals. (**C**) Metagene analysis comparing the average Hi-C signal (detecting long-range chromatin interactions) for strongly CDK9-responding TUs (red, response ratio >75% quantile, 552 TUs) and weakly CDK9-responding TUs (black, response ratio <25% quantile, 440 TUs) aligned at the pause site. Shaded areas around the average signal (solid lines) indicate confidence intervals (Materials and methods, ***Supplementary file 1***).

DOI: https://doi.org/10.7554/eLife.29736.015

The following figure supplement is available for figure 7:

**Figure supplement 1.** Features of promoter-proximal pausing.

DOI: https://doi.org/10.7554/eLife.29736.016

previous report (***Hendrix et al., 2008***). Comparison of strongly and weakly CDK9-responding TUs around the pause site showed that TUs that responded strongly to CDK9 inhibition showed a higher tendency to establish long-range chromatin interactions (***Figure 7C***) as observed by Hi-C (***Ma et al., 2015***). This is consistent with the idea that interactions of an enhancer with its target promoter can stimulate Pol II pause release (***Ghavi-Helm et al., 2014***; ***Rahl et al., 2010***). This tendency however seems to be independent of the pause duration as comparing TUs with long and short pause durations leads to no observable difference in Hi-C signal.

Finally, we investigated which factors preferentially occupy pause windows with longer pause durations. This is now possible because ChIP-seq signals can be normalized with the productive initiation frequency. Without such normalization, ChIP-seq derived factor occupancies are artificially high in pause windows with long pause durations (***Ehrensberger et al., 2013***). Correlation of such

normalized ChIP-seq signals in the pause window with pause durations (*Figure 7—figure supplement 1B–C*) resulted in a positive correlation for Pol II phosphorylation at sites that are associated with elongation, and also for NELF-E, CDK9, and Brd4, which are all factors involved in Pol II pausing and release.

## Discussion

Taken together, our results show that Pol II pausing can control transcription initiation and demonstrate the central role of CDK9 in controlling pause duration and thereby the productive initiation frequency. Our results have implications for understanding gene regulation. Genes that show initiation frequencies below the pause-initiation limit may be activated by increasing the initiation frequency without changing pause duration. However, activation of genes that are transcribed at the pause-initiation limit requires a decrease in pause duration, that is stimulation of pause release, to enable higher initiation frequencies. We suggest that pause-controlled initiation evolved because mutations in the promoter-proximal region can change pause duration, and thereby limit initiation, but do not compromise a high initiation capacity of the core promoter around the TSS. This may have enabled the evolution of genes that remain highly inducible but can be efficiently downregulated.

After our work had been completed, a publication appeared that concluded that polymerase pausing inhibits new transcription initiation (*Shao and Zeitlinger, 2017*). The conclusion in this paper is consistent with our general finding of an interdependency of Pol II pausing and transcription initiation, but the two studies differ in three aspects. First, we used human cells whereas the published work was conducted in Drosophila cells. Second, our work uses a multi-omics approach to enable a kinetic description, whereas the published work is based on changes in factor occupancy. Third, we selectively inhibited CDK9 using CRISPR-Cas9-based engineering and chemical biology, whereas the published work used small molecule inhibitors that may target multiple kinases. Despite these differences, the general conclusion that promoter-proximal pausing of Pol II sets a limit to the frequency of transcription initiation holds for both human and Drosophila cells and is likely a general feature of metazoan gene regulation.

## Materials and methods

### Key resources table

| Reagent type (species) or resource | Designation | Source or reference | Identifiers | Additional information |
|---|---|---|---|---|
| cell line (Homo sapiens; male) | Raji B lymphocyte cells (wild type) | DSMZ | DSMZ Cat# ACC-319; RRID:CVCL_0511 | |
| cell line (Homo sapiens; male) | Raji B lymphocyte cells (CDK9[as]) | This paper | | Raji B cells were obtained from DSMZ Cat# ACC-319, RRID:CVCL_0511. Homozygous mutation of F103 at the CDK9 gene loci in Raji B cells was performed using the CRISPR-Cas9 system. |
| antibody | anti-CDK9 | Santa Cruz, Dallas, TX USA | sc-484 | |
| antibody | anti-alpha-tubulin | Sigma-Aldrich, St. Louis, MO USA | DM1A | |
| antibody | anti-Pol II (total, unphos + phos) | BIOZOL, Eching, Germany | MABI0601 | |
| commercial assay or kit | CellTiter 96 AQueous One Solution Cell Proliferation Assay (MTS) | Promega, Madison, WI USA | G3582 | |
| commercial assay or kit | Plasmo Test Mycoplasma Detection Kit | InvivoGen, San Diego, CA USA | rep-pt1 | |

*Continued on next page*

*Continued*

| Reagent type (species) or resource | Designation | Source or reference | Identifiers | Additional information |
|---|---|---|---|---|
| commercial assay or kit | Ovation Universal RNA-Seq System | NuGEN, Leek, The Netherlands | 0343–32 | |
| commercial assay or kit | TruSeq Small RNA Library Prep Kit | Illumina, Massachusetts USA | RS-200–0012 | |
| chemical compound, drug | CDK9as inhibitor; 1-NA-PP1 | Calbiochem, EMD Millipore, Danvers, MA USA | 529579 | CAS 221243-82-9 |
| chemical compound, drug | Solvent control; DMSO | Sigma-Aldrich, St. Louis, MO USA | D8418 | |
| chemical compound, drug | 4-thiouracil (4sU) | Sigma-Aldrich, St. Louis, MO USA | T4509 | |
| chemical compound, drug | empigen BB detergent | Sigma-Aldrich, St. Louis, MO USA | 30326 | |

## Cell lines and cell culture

Raji B cells were obtained from DSMZ (DSMZ no.: ACC 319; RRID:CVCL_0511). CDK9[as] Raji B cells were generated in this study by CRISPR-Cas9-based engineering of Raji B cells obtained from DSMZ (DSMZ no.: ACC 319; RRID:CVCL_0511). Raji B cells and CDK9[as] Raji B cells were grown in RPMI 1640 medium (Thermo Fisher Scientific, Waltham, MA USA) supplemented with 10% foetal calf serum (bio-sell, Nürnberg, Germany), 100 U/mL penicillin and 100 µg/mL streptomycin (Thermo Fisher Scientific, Waltham, MA USA), and 2 mM L-glutamine (Thermo Fisher Scientific, Waltham, MA USA) at 37°C and 5% $CO_2$. Cells were verified to be free of mycoplasma contamination using Plasmo Test Mycoplasma Detection Kit (InvivoGen, San Diego, CA USA).

## Generation of human CDK9[as]Raji B cell line

CDK9[as] contains a point mutation of the so-called gatekeeper residue that enables the kinase active site to accept bulky ATP analogs (1-NA-PP1) (4-Amino-1-tert-butyl-3-(1'-naphthyl)pyrazolo[3,4-d] pyrimidine). To identify the gatekeeper residue (*Lopez et al., 2014*), the amino acid sequence of the human CDK9 kinase (UniProt, P50750-1) was aligned with sequences of previously characterized kinases carrying analog sensitive mutations. Multiple sequence alignment was performed with the web tool Clustal Omega 1.2.4 (*Sievers et al., 2011*). For the canonical isoform of CDK9, phenylalanine (F) 103 was identified as the gatekeeper residue and selected for mutation to alanine (A). Mutation of F103 at the CDK9 gene loci in Raji B cells was performed using the CRISPR-Cas9 system (*Doudna and Charpentier, 2014*; *Hsu et al., 2014*) as described (*Mulholland et al., 2015*) with minor modifications. Briefly, the single guide RNA (sgRNA) for editing CDK9 was designed by using the web tool Optimized CRISPR design (http://crispr.mit.edu/), and was incorporated to pSpCas9 (BB)−2A-GFP (PX458) vector by BpiI restriction sites (Addgene plasmid # 48138) (*Ran et al., 2013*). For nucleotide replacement (gttc to cgcg), 200 nt single-stranded DNA oligonucleotides (ssODNs) were synthesized by Integrated DNA Technologies (IDT, Leuven, Belgium) and used as homology-directed repair (HDR) template. A BstUI cutting site was incorporated into the HDR template for screening. The vector and HDR template were introduced into human Raji B cells using Amaxa Mouse ES Cell Nucleofector Kit (Lonza, Basel, Switzerland) according to the manufacturer's instructions. Two days after transfection, GFP positive cells were single cell sorted into 96 well plates using FACS Aria II instrument (Becton Dickinson, Franklin Lakes, NJ USA). After two weeks, individual colonies were expanded for genomic DNA isolation. The mutant lines were validated by PCR using respective primers, BstUI digestion (*Figure 1—figure supplement 2A*) and DNA sequencing (*Figure 1—figure supplement 2B*).

HDR template (A103 is underlined, BstUI cutting site in small letters):

AAAGTGTGTTGGGTGTGGTTTTCTTGACTTTTTCTTCTTTCTATTCCTGCCTCAGCTTCCCCCTA TAACCGCTGCAAGGGTAGTATATACCTGGTcgcgGACTTCTGCGAGCATGACCTTGCTGGGCTG TTGAGCAATGTTTTGGTCAAGTTCACGCTGTCTGAGATCAAGAGGGTGATGCAGATGCTGC TTAACGGCCT

Primers for sgRNA generation and screening:
CDK9-sgRNA-F: 5'-CACCGGCTCGCAGAAGTCGAACACC-3'
CDK9-sgRNA-R: 5'-AAACGGTGTTCGACTTCTGCGAGCC-3'
CDK9-screen-F: 5'-CCCCGTAGCTGGTGCTTCCTCG-3'
CDK9-screen-R: 5'-CCCCAGCAGCCTTCATGTCCCTAT-3'

## Antibodies and western blot analysis

Proteins equivalent to $1 \times 10^5$ Raji B cells were loaded in Laemmli buffer and subjected to SDS-PAGE before transfer to nitrocellulose. Unspecific binding of antibodies was blocked by incubation of the membrane with 5% milk in Tris-buffered saline containing 1% Tween. Primary antibodies were anti-CDK9 (sc-484) (Santa Cruz, Dallas, TX USA) and anti-α-tubulin (DM1A) (Sigma-Aldrich, St. Louis, MO USA). Fluorophore-coupled secondary antibodies (Rockland Immunochemicals Inc., Pottstown, PA USA) were used and blots were visualized using the Odyssey system (LI-COR, Lincoln, NE USA).

## MTS assay

Cell proliferation at increasing 1-NA-PP1 inhibitor concentrations was measured in four biological replicates using the CellTiter 96 AQueous One Solution Cell Proliferation Assay System (Promega, Madison, WI USA). Cells were seeded in a 96-well plate and increasing concentrations of 1-NA-PP1 (Calbiochem, EMD Millipore, Danvers, MA USA) or DMSO (Sigma-Aldrich, St. Louis, MO USA) were added. After 72 hr, MTS tetrazolium compound was added to each well for one hour. Subsequently, the quantity of the MTS formazan product was measured as absorbance at 490 nm with a Sunrise photometer (TECAN, Männedorf, Switzerland) that was operated using the Magellan data analysis software (v7.2, TECAN, Männedorf, Switzerland). Relative signals for each concentration were calculated by dividing the signals of the CDK9$^{as}$ inhibitor treated cells by the corresponding signals of the control.

## TT-seq

Two biological replicates of reactions including RNA spike-ins were performed essentially as described (*Schwalb et al., 2016*). Briefly, $3.3 \times 10^7$ Raji B (CDK9$^{as}$ or wild type) cells were treated for 15 min with solvent DMSO (control) or 5 μM of 1-NA-PP1 (CDK9$^{as}$ inhibitor). After 10 min of treatment, labeling was performed by adding 500 μM of 4-thiouracil (4sU) (Sigma-Aldrich, St. Louis, MO, USA) for 5 min at 37°C and 5% $CO_2$. Cells were harvested by centrifugation at 3000 x g for 2 min. Total RNA was extracted using QIAzol according to the manufacturer's instructions. RNAs were sonicated to generate fragments of <1.5 kbp using AFAmicro tubes in a S220 Focused-ultrasonicator (Covaris Inc., Woburn, MA USA). 4sU-labeled RNA was purified from 150 μg total fragmented RNA. Separation of labeled RNA was achieved with streptavidin beads (Miltenyi Biotec, Bergisch Gladbach, Germany) as described in (*Schwalb et al., 2016*). Prior to library preparation, 4sU-labeled RNA was purified and quantified. Enrichment of 4sU-labeled RNA was analyzed by RT-qPCR as described (*Schwalb et al., 2016*). Input RNA was treated with HL-dsDNase (ArcticZymes, Tromsø, Norway) and used for strand-specific library preparation according to the Ovation Universal RNA-Seq System (NuGEN, Leek, The Netherlands). The size-selected and pre-amplified fragments were analyzed on a Fragment Analyzer before clustering and sequencing on the Illumina HiSeq 1500.

## TT-seq data preprocessing and global normalization

Paired-end 50 base reads with additional 6 base reads of barcodes were obtained for each of the samples, that is two TT-seq replicates with 1-NA-PP1 (CDK9$^{as}$ inhibitor) and two TT-Seq replicates with DMSO (control) treatment. Reads were demultiplexed and mapped with STAR 2.3.0 (*Dobin and Gingeras, 2015*) to the hg20/hg38 (GRCh38) genome assembly (Human Genome Reference Consortium). Samtools (*Li et al., 2009*) was used to quality filter SAM files, whereby alignments with MAPQ smaller than 7 (-q 7) were skipped and only proper pairs (-f2) were selected. Further data processing was carried out using the R/Bioconductor environment. We used a spike-in (RNAs) normalization strategy essentially as described (*Schwalb et al., 2016*) to allow observation of global shifts and antisense bias determination (ratio of spurious reads originating from the opposite strand introduced by the RT reactions). Read counts for spike-ins were calculated using HTSeq

(*Anders et al., 2015*). Sequencing depth calculations did not detect global differences. Antisense bias ratios were calculated for each sample $j$ according to

$$c_j = \underset{i}{\mathrm{median}} \left( \frac{k_{ij}^{antisense}}{k_{ij}^{sense}} \right)$$

for all available spike-ins $i$.

## Definition of transcription units (TUs)

For each annotated gene, transcription units (TUs) were defined as the union of all existing inherent transcript isoforms (UCSC RefSeq GRCh38). Read counts for all features were calculated using HTSeq (*Anders et al., 2015*) and corrected for antisense bias using antisense bias ratios $c_j$ calculated as described above. The real number of read counts $s_{ij}$ for transcribed unit $i$ in sample $j$ was calculated as

$$s_{ij} = \frac{S_{ij} - c_j A_{ij}}{1 - c_j^2}$$

where $S_{ij}$ and $A_{ij}$ are the observed number of read counts on the sense and antisense strand. Read counts per kilobase (RPK) were calculated upon bias corrected read counts falling into the region of a transcribed unit divided by it's length in kilobases. Based on the antisense bias corrected RPKs a subgroup of expressed TUs was defined to comprise all TUs with an RPK of 100 or higher in two summarized replicates of TT-seq without inhibitor treatment. An RPK of 100 corresponds to approximately a coverage of 10 per sample due to an average fragment size of 200. This subset was used throughout the analysis unless stated otherwise.

## Calculation of the number of transcribed bases

Aligned duplicated fragments were discarded for each sample. Of the resulting unique fragment isoforms only those were kept that exhibited a positive inner mate distance. The number of transcribed bases ($tb_j$) for all samples was calculated as the sum of the coverage of evident (sequenced) fragment parts (read pairs only) for all fragments smaller than 500 bases in length and with an inner mate interval not entirely overlapping a Refseq annotated intron (UCSC RefSeq GRCh38, ~96% of all fragments) in addition to the sum of the coverage of non-evident fragment parts (entire fragment).

## Size factor normalization

We first checked that no significant global shifts were detected in a comparison of two TT-seq replicates with 1-NA-PP1 (CDK9[as] inhibitor) treatment against two TT-seq replicates with DMSO treatment (control) in the above described spike-ins normalization strategy. Then all samples were subjected to an alternative, more robust normalization procedure. For each sample $j$ the antisense bias corrected number of transcribed bases $tb_j$ was calculated on all expressed TUs $i$ exceeding 125 kbp in length. 50 kbp were truncated from each side of the selected TUs to avoid influence of the response to CDK9[as] inhibition (*Laitem et al., 2015*). On the resulting intervals, size factors for each sample j were determined as

$$\sigma_j = \underset{i}{\mathrm{median}} \left( \frac{tb_{ij}}{\left( \prod_{v=1}^{m} tb_{ij} \right)^{1/m}} \right)$$

where $m$ denotes the number of samples. This formula has been adapted (*Anders and Huber, 2010*) and was used to correct for library size and sequencing depth variations.

## Calculation of response ratios

For each condition j (control or CDK9[as] inhibited) the antisense bias corrected number of transcribed bases $tb_i^j$ was calculated on all expressed TUs $i$ exceeding 10 kbp in length. Of all remaining TUs only those were kept harboring one unique TSS given all Refseq annotated isoforms (UCSC RefSeq GRCh38). Response ratios were calculated for a window from the TSS to 10 kbp downstream (excluding the first 200 bp) for each TU $i$ as

$$r_i = 1 - tb_{i\,[0.2\,,\,10\,kbp]}^{CDK9^{as}\ inhibited} / tb_{i\,[0.2\,,\,10\,kbp]}^{Control}$$

where negative values were set to 0.

## Estimation of robust common elongation velocity

For each condition j (control or CDK9[as] inhibited) the antisense bias corrected number of transcribed bases $tb_i^j$ was calculated on all expressed TUs $i$ with a given response ratio $r_i$, excluding the first 200 bp. All TUs were truncated by 5 kbp in length from the 3' end prior to calculation to avoid influence of some alterations in signal around the pA site after CDK9[as] inhibition (*Laitem et al., 2015*). A robust common elongation velocity estimate was calculated by finding an optimal fit for all TUs $i$ between 25 to 200 kbp in length $L_i$, that is minimizing the function

$$loss = \underset{i}{median}\left( \left| 1 - \frac{tb_i^{CDK9^{as}\ inhibited}}{tb_i^{Control}} - \frac{r_i v (t^* - t)}{L_i} \right| \right)$$

on the interval [0,10000] with inhibitor treatment duration $t^*$=15 [min] and labeling duration $t$ = 5 [min], given that

$$tb_i^{CDK9^{as}\ inhibited} - tb_i^{Control} = r_i \frac{tb_i^{Control}}{L_i} v_i (t^* - t)$$

that is the difference of transcribed bases obtained by the CDK9[as] inhibitor treatment equals the number of transcribed bases per nucleotide $tb_i^{Control}/L_i$ times the number of nucleotides traveled $v_i(t^* - t)$ in $t^* - t$ minutes corrected by the amount of the response $r_i$.

## Estimation of gene-wise elongation velocity

For each condition $j$ (control or CDK9[as] inhibited) the antisense bias corrected number of transcribed bases $tb_i^j$ was calculated on all expressed TUs $i$ exceeding 35 kbp in length, excluding the first 200 bp. All TUs were truncated by 5 kbp in length from the 3' end prior to calculation to avoid influence of some alterations in signal around the pA site after CDK9[as] inhibition (*Laitem et al., 2015*). Of all remaining TUs only those were kept harboring one unique TSS given all Refseq annotated isoforms (UCSC RefSeq GRCh38). For each TU $i$ with $r_i$>0.25 the elongation velocity $v_i$ [kbp/min] was calculated as

$$v_i = \frac{tb_i^{Control} - tb_i^{CDK9^{as}\ inhibited}}{tb_i^{Control} \cdot \frac{r_i}{L_i} (t^* - t)}$$

with inhibitor treatment duration $t^*$=15 [min] and labeling duration $t$ = 5 [min].

## mNET-seq

Two biological replicates of reactions including empigen BB detergent treatment during immuno-precipitation (IP) were performed essentially as described (*Nojima et al., 2016*; *Schlackow et al., 2017*), with minor modifications. Briefly, $1.6 \times 10^8$ Raji B (CDK9[as]) cells were treated for 15 min with solvent DMSO (control) or 5 µM of 1-NA-PP1 (CDK9[as] inhibitor). Cell fractionation was performed as described (*Conrad and Ørom, 2017*). Isolated chromatin was digested with micrococcal nuclease (MNase) (NEB, Ipswich, MA USA) at 37°C and 1,400 rpm for 90 s. To inactivate MNase, EGTA was added to a final concentration of 25 mM. Digested chromatin was collected by centrifugation at 4°C and 13,000 rpm for 5 min. The supernatant was diluted tenfold with IP buffer containing 50 mM Tris-HCl pH 7.5, 150 mM NaCl, 0.05% (vol/vol) NP-40, and 1% (vol/vol) empigen BB (Sigma-Aldrich, St. Louis, MO USA). For each IP, 50 µg of Pol II antibody clone MABI0601 (BIOZOL, Eching, Germany) was conjugated to Dynabeads M-280 Sheep Anti-Mouse IgG (Thermo Fisher Scientific, Waltham, MA USA). Pol II antibody-conjugated beads were added to diluted sample. IP was performed on a rotating wheel at 4°C for 1 hr. The beads were washed six times with IP buffer (50 mM Tris-HCl pH 7.5, 150 mM NaCl, 0.05 % NP-40, and 1% empigen BB) and once with 500 µL of PNKT buffer containing 1 x T4 polynucleotide kinase (PNK) buffer (NEB, Ipswich, MA USA) and 0.1% (vol/vol) Tween-20 (Sigma-Aldrich, St. Louis, MO USA). Beads were incubated in 100 µL of PNK reaction mix

containing 1 x PNK buffer, 0.1% (vol/vol) Tween-20, 1 mM ATP, and T4 PNK, 3' phosphatase minus (NEB, Ipswich, MA USA) at 37°C for 10 min. Beads were washed once with IP buffer. RNA was extracted with TRIzol reagent. RNA was precipitated with GlycoBlue co-precipitant (Thermo Fisher Scientific, Waltham, MA USA) and resolved on 6% denaturing acrylamide containing 7 M urea (Pan-Reac AppliChem, Darmstadt, Germany) gel for size purification. Fragments of 35–100 nt were eluted from the gel using elution buffer containing 1 M NaOAc, 1 mM EDTA, and precipitated in ethanol. RNA libraries were prepared according to the TruSeq Small RNA Library Kit (Illumina, Massachusetts USA) and as described (*Nojima et al., 2016*). The size-selected and pre-amplified fragments were analyzed on a Fragment Analyzer before clustering and sequencing on an Illumina HiSeq 2500 sequencer.

## mNET-seq data preprocessing and normalization

Paired-end 50 base reads with additional 6 base reads of barcodes were obtained for each of the samples, that is mNET-seq samples with 1-NA-PP1 (CDK9$^{as}$ inhibitor) and with DMSO (control) treatment. Reads were demultiplexed and mapped with STAR 2.3.0 (*Dobin and Gingeras, 2015*) to the hg20/hg38 (GRCh38) genome assembly (Human Genome Reference Consortium). Samtools (*Li et al., 2009*) was used to quality filter SAM files, whereby alignments with MAPQ smaller than 7 (-q 7) were skipped and only proper pairs (-f2) were selected. Further data processing was carried out using the R/Bioconductor environment. Antisense bias (ratio of spurious reads originating from the opposite strand introduced by the RT reactions) was determined using positions in regions without antisense annotation with a coverage of at least 100 according to Refseq annotated genes (UCSC RefSeq GRCh38). mNET-seq coverage tracks were size factor normalized on 260 TUs that showed a response of less than 5% ($r_i < 0.05$) in the TT-seq signal upon 1-NA-PP1 (CDK9$^{as}$ inhibitor) treatment. The response ratio $r_i$ was determined as described above including also TUs with multiple TSS to extend the number of TUs for normalization. Note that variation of the response ratio cutoff and thereby the number of TUs available for normalization does virtually not change the normalization parameters. Coverage tracks for further analysis were restricted to the last nucleotide incorporated by the polymerase in the aligned mNET-seq reads.

## Detection of pause sites

For all expressed TUs *i* exceeding 10 kbp in length with one unique TSS given all Refseq annotated isoforms (UCSC RefSeq GRCh38) the pause site *m** was calculated for all bases *m* in a window from the TSS to the end of the first exon (excluding the last 5 bases) via maximizing the function

$$\rho_i = \max_m p_{im}$$

where $\rho_i$ needed to exceed 5 times the median of the signal strength $p_{im}$ for all non-negative antisense bias corrected mNET-seq coverage values (*Nojima et al., 2015*). Note that all provided coverage tracks were used.

## DNA-RNA and DNA-DNA melting temperature calculation

The gene-wise mean melting temperature of the DNA-RNA and DNA-DNA hybrid was calculated from subsequent melting temperature estimates of 8-base pair DNA-RNA and DNA-DNA duplexes tiling the respective area according to (*SantaLucia, 1998*; *Sugimoto et al., 1995*).

## Molecular weight conversions

The known sequence and mixture of the utilized spike-ins allows to calculate a conversion factor to RNA amount per cell [cell$^{-1}$] given their molecular weight assuming perfect RNA extraction. The number of spike-in molecules per cell $N$ [cell$^{-1}$] was calculated as

$$N = \frac{m}{Mn} N_A$$

with the number of spike-ins $m$ $25.10^{-9}$ [g], the number of cells $n$ $3.27.10^7$, the Avogadro constant $N_A$ $6.02214085774.10^{23}$ [mol$^{-1}$] and molar-mass (molecular weight) of the spike-ins $M$ [g mol$^{-1}$] calculated as

$$M = A_n \cdot 329.2 + (1 - \tau) \cdot U_n \cdot 306.2 + C_n \cdot 305.2 + G_n \cdot 345.2 + \tau \cdot 4sU_n \cdot 322.26 + 159$$

where $A_n$, $U_n$, $C_n$, $G_n$ and $4sU_n$ are the number of each respective nucleotide within each spike-in polynucleotide. $\tau$ - 1 is set to 0.1 in case of a labeled spike-in and 0 otherwise. The addition of 159 to the molecular weight takes into account the molecular weight of a 5' triphosphate. Provided the above the conversion factor to RNA amount per cell $\kappa$ - 1 [cell$^{-1}$] can be calculated as

$$\kappa = \text{mean}\left( \text{median}_i \left( \frac{tb_i}{L_i \cdot N} \right) \right)$$

for all labeled spike-in species $i$ with length $L_i$. Note that imperfect RNA extraction efficiency would lead to an underestimation of cellular labeled RNA in comparison to the amount of added spike-ins and thus to an underestimation of initiation frequencies. In case of a strong underestimation however the real initiation frequencies would lie above the pause-initiation limit, which is theoretically impossible. Thus we assume this effect to be insignificant.

## Estimation of initiation frequency *I*

The antisense bias corrected number of transcribed bases $tb_i^{Control}$ was calculated on all expressed TUs $i$ exceeding 10 kbp in length. Of all remaining TUs only those were kept harboring one unique TSS given all Refseq annotated isoforms (UCSC RefSeq GRCh38). For each TU $i$ the productive initiation frequency $I_i$ [cell$^{-1}$min$^{-1}$], which corresponds to the pause release rate, was calculated as

$$I_i = \frac{1}{\kappa} \cdot \frac{tb_i^{Control}}{t \cdot L_i}$$

with labeling duration $t$ = 5 [min] and length $L_i$. Note that $tb_i^{Control}$ and $L_i$ were restricted to regions of non-first constitutive exons (exonic bases common to all isoforms).

## Estimation of pause duration *d*

For all expressed TUs $i$ exceeding 10 kbp in length with one unique TSS given all Refseq annotated isoforms (UCSC RefSeq GRCh38) the pause duration $d_i$ [min] was calculated as the residing time of the polymerase in a window ±100 bases $m$ around the pause site (see above) as

$$d_i = \frac{\sum_{+/-100} p_{im}}{I_i} \cdot \text{median}_i \left( \frac{v_i}{I_i v_i (t^* - t) / \sum_{response\ window} p_{im}} \right)$$

with pause release rate $I_i$ and the number of polymerases $p_{im}$ (antisense bias corrected mNET-seq coverage values [*Nojima et al., 2015*]) in a window ±100 bases around the pause site. For pause sites below 100 bp downstream of the TSS the first 200 bp of the TU were considered. Note that the right part of the formula is restricted to mNETseq instances above the 50% quantile for robustness and adjusts $d_i$ to an absolute scale by comparing the CDK9 derived elongation velocities $v_i$ with those derived from combining mNET-seq and TT-seq data in the response window $[200, v_i(t^* - t)]$.

## Pause-initiation limit

The previously derived inequality from (*Ehrensberger et al., 2013*)

$$\frac{v}{I} \geq 50\ [bp]$$

states that new initiation events into productive elongation are limited by the velocity of the polymerase in the promoter-proximal region and that steric hindrance occurs below a distance of 50 bp between the active sites of the initiating Pol II and the paused Pol II. Given the calculations of pause duration $d$ and (productive) initiation frequency $I$ above, we can reformulate this inequality to

$$\frac{200\ [bp]}{d \cdot I} \geq 50\ [bp]$$

with 200 [bp] being the above defined pause window.

## Simulation of TT-seq data based on elongation velocity profiles

Based on the following model we simulated TT-seq coverage values by providing elongation velocity profiles $v(t)$, a labeling duration $t^{lab}$ and a uracil content dependent labeling bias

$$l_f = 1 - \left(1 - p^{lab}\right)^{\#u_f}$$

$p^{lab}$ denotes the labeling probability (set to 0.05) and $\#u_f$ the number of uracil residues of a given fragment $f$ (set to 0.28 times fragment length). The elongation velocity profile $v(t)$ can be used to calculate the number of elongated positions of the polymerase $\tau(t)$ at timepoint $t$ as

$$\tau(t) = \int_0^t v(t)dt$$

Given the transcription start site $\tau(0)$ the number of elongated positions $\tau(t)$ can be used to determine the end of an emerging nascent fragment $f$. Based on that we determined the start position of a fragment as $\tau\left(\max(t - t^{lab}, 0)\right)$ for each labeling duration $t^{lab}$ as the position of the polymerase at the beginning of the labeling process. Subsequently, we used the number of uracil residues present in the RNA fragment $\#u_f$ to weight the amount of coverage contributed by this fragment as $l_f$. Additionally, we applied a size selection similar to that in the original protocol for fragments below 80 bp in length with a sigmoidal curve that mimics a typical size selection spread. Given a pause position of 80 bp downstream of the TSS and pause duration of 1 or 2 min we adjusted the elongation velocity profile to simulate polymerase pausing. Note that neither reasonable changes in labeling probability, size selection probability nor changes in uracil residue content change the general observation that longer pause durations induce a greater shortage of TT-seq coverage in the region between the TSS and the pause site.

## Estimation of gene-wise elongation velocity (without of response ratio)

For each condition $j$ (control or CDK9[as] inhibited) the antisense bias corrected number of transcribed bases $tb_i^j$ was calculated on all expressed TUs $i$ exceeding 35 kbp in length, excluding the first 200 bp. All TUs were truncated by 5 kb in length from the 3′ end prior to calculation to avoid influence of some alterations in signal around the pA site after CDK9[as] inhibition (*Laitem et al., 2015*). Of all remaining TUs only those were kept harboring one unique TSS given all Refseq annotated isoforms (UCSC RefSeq GRCh38). For each TU $i$ with $r_i > 0.25$ the cumulative sums of the difference of the number of transcribed bases $tb_i^j$ for each base k was calculated as

$$S_0 = 0 \quad S_n = S_{n-1} + tb_i^{Control} - tb_i^{CDK9^{as}\ inhibited}$$

starting at the unique TSS (position 0) to $n = L_i$ the length of the TU. A elongation length estimate $L_i^{response\ window}$ was then calculated by finding an optimal fit for n between 0 to $L_i$, that is maximizing the function

$$gain = \max_n \left( \frac{S_n \cdot L_i}{\max\limits_{n=1..L_i} S_n} - n + 1 \right)$$

on the interval $[0, L_i]$. In words, finding the maximum of the cumulative sums of difference in coverage rotated 45 degrees clockwise. The elongation velocity $\hat{v}_i$ [kbp/min] was subsequently calculated as

$$\hat{v}_i = \frac{L_i^{response\ window}}{(t^* - t)}$$

with inhibitor treatment duration $t^* = 15$ [min] and labeling duration $t = 5$ [min].

### Estimation of pause duration $\hat{d}$ (without of initiation frequency).

For all expressed TUs $i$ exceeding 10 kb in length with one unique TSS given all Refseq annotated isoforms (UCSC RefSeq GRCh38) the pause duration $\hat{d}_i$ [min] was calculated as the residing time of the polymerase in a window ±100 bases $m$ around the pause site (see above) as

$$\hat{d}_i = \frac{\sum_{+/-100} p_{im} \cdot L_i^{response\ window}}{\sum_{response\ window} p_{im} \cdot \hat{v}_i}$$

with elongation length estimate $L_i^{response\ window}$ and the number of polymerases $p_{im}$ (antisense bias corrected mNET-seq coverage values) in a window ±100 bases around the pause site. For pause sites below 100 bp downstream of the TSS the first 200 bp of the TU were considered. Note that $\hat{d}_i$ was adjusted to the height as $d_i$ by a single proportionality factor for visualization purposes.

### In vivo RNA secondary structure (DMS-seq [*Rouskin et al., 2014*])

The gene-wise DMS-seq coverage (300 µl in vivo) for a window of [−15,–65] bp upstream of the pause site was normalized by subtraction from the respective DMS-seq coverage (denatured) allowing for maximal 5% negative values which were set to 0 (sequencing depth adjustment). The gene-wise mean values were subsequently normalized by dividing with the initiation frequency. Note that the latter normalization has an insignificant effect.

### Prediction of RNA secondary structure

The gene-wise mean minimum free energy for a window of [−15,–65] bp upstream of the pause site was calculated from subsequent minimum free energy estimates of 13-base pair RNA fragments tiling the respective area using RNAfold from the ViennaRNA package (*Lorenz et al., 2011*).

## Acknowledgements

We would like to thank Helmut Blum and Stefan Krebs (LAFUGA, LMU Munich) for sequencing. We also thank Merle Hantsche for structural modeling. We thank Julien Gagneur (Technical University of Munich) for initial discussions. HL was funded by SFB 1064 TP A17. DE was funded by SFB1064 (Chromatin Dynamics). PC was funded by Advanced Grant TRANSREGULON of the European Research Council and the Volkswagen Foundation.

## Additional information

### Funding

| Funder | Grant reference number | Author |
|---|---|---|
| European Research Council | TRANSREGULON | Patrick Cramer |
| Volkswagen Foundation | | Patrick Cramer |
| Deutsche Forschungsgemeinschaft | SFB 1064 TP A17 | Heinrich Leonhardt |
| Deutsche Forschungsgemeinschaft | SFB 1064 | Dirk Eick |
| Max Planck Institute for Biophysical Chemistry | Open-access funding | Patrick Cramer |

The funders had no role in study design, data collection and interpretation, or the decision to submit the work for publication.

### Author contributions

Saskia Gressel, Data curation, Validation, Investigation, Visualization, Methodology, Writing—original draft, Writing—review and editing, Optimized and carried out TT-seq and mNET-seq experiments and contributed to the design of bioinformatics analysis, Prepared the manuscript, with input from

all authors; Björn Schwalb, Conceptualization, Data curation, Software, Formal analysis, Supervision, Validation, Investigation, Visualization, Methodology, Writing—original draft, Writing—review and editing, Designed and carried out bioinformatics analysis, Designed and supervised research, Prepared the manuscript, with input from all authors; Tim Michael Decker, Validation, Visualization, Methodology, Carried out cellular and biochemical characterization of the CDK9as strain and contributed to TT-seq experiments of CDK9as strain; Weihua Qin, Validation, Visualization, Methodology, Generated and validated the CDK9as cell line; Heinrich Leonhardt, Resources, Supervision, Funding acquisition, Validation, Methodology, Designed and supervised CDK9as strain generation, validation and characterization; Dirk Eick, Conceptualization, Resources, Supervision, Funding acquisition, Validation, Methodology, DE designed and supervised CDK9as strain generation, validation and characterization; Patrick Cramer, Conceptualization, Resources, Supervision, Funding acquisition, Investigation, Visualization, Methodology, Writing—original draft, Project administration, Writing—review and editing, Designed and supervised research, Prepared the manuscript, with input from all authors

### Author ORCIDs
Saskia Gressel (iD) https://orcid.org/0000-0003-0261-675X
Björn Schwalb (iD) https://orcid.org/0000-0003-2987-2622
Patrick Cramer (iD) https://orcid.org/0000-0001-5454-7755

### Decision letter and Author response
Decision letter https://doi.org/10.7554/eLife.29736.043
Author response https://doi.org/10.7554/eLife.29736.044

## Additional files

### Supplementary files
• Supplementary file 1. Published datasets used for analysis. Note that the conclusions we draw across different cell-lines are all based on metagene analysis, involving from 500 up to more than 2000 genes. Thus, we assume cell-line specific differences to have an insignificant influence and that the tendencies we observe rather suggest strong conservation.
DOI: https://doi.org/10.7554/eLife.29736.018

• Transparent reporting form
DOI: https://doi.org/10.7554/eLife.29736.019

### Major datasets
The following dataset was generated:

| Author(s) | Year | Dataset title | Dataset URL | Database, license, and accessibility information |
|---|---|---|---|---|
| Gressel S, Schwalb B, Decker TM, Qin W, Leonhardt H, Eick D, Cramer P | 2017 | CDK9-dependent RNA polymerase II pausing controls transcription initiation | https://www.ncbi.nlm.nih.gov/geo/query/acc.cgi?acc=GSE96056 | Publicly available at the NCBI Gene Expression Omnibus (accession no: GSE96056) |

The following previously published datasets were used:

| Author(s) | Year | Dataset title | Dataset URL | Database, license, and accessibility information |
|---|---|---|---|---|
| Furey T, Boyle A, Song L, Crawford G, Giresi P, Lieb J, Liu Z, McDaniell R, Lee B, Iyer V, Flicek P, Keefe D, Birney E, Graf S | 2012 | Open Chromatin by FAIRE from ENCODE/OpenChrom(UNC Chapel Hill) | https://www.ncbi.nlm.nih.gov/geo/query/acc.cgi?acc=GSE35239 | Publicly available at the NCBI Gene Expression Omnibus (accession no: GSE35239) |

| | | | | |
|---|---|---|---|---|
| Ma W, Ay F, Lee C, Gulsoy G, Deng X, Cook S, Hesson J, Ware CB, Krumm A, Shendure J, Blau CA, Disteche C, Noble WS, Duan Z | 2014 | Fine-scale chromatin interaction maps reveal the cis-regulatory landscape of human lincRNA genes | https://www.ncbi.nlm.nih.gov/geo/query/acc.cgi?acc=GSE56869 | Publicly available at the NCBI Gene Expression Omnibus (accession no: GSE56869) |
| Varley K, Gertz J, Myers RM | 2011 | DNA Methylation by Reduced Representation Bisulfite Seq from ENCODE/HudsonAlpha | https://www.ncbi.nlm.nih.gov/geo/query/acc.cgi?acc=GSE27584 | Publicly available at the NCBI Gene Expression Omnibus (accession no: GSE27584) |
| Baranello L, Wojtowicz D, Kouzine F, Cui K, Chan-Salis KY, Devaiah BN, Singer D, Pommier Y, Pugh BF, Przytycka TM, Lewis BA, Zhao K, Levens D | 2016 | Study of Topoisomerase I in human | https://www.ncbi.nlm.nih.gov/geo/query/acc.cgi?acc=GSE57628 | Publicly available at the NCBI Gene Expression Omnibus (accession no: GSE57628) |
| Furey T, Boyle A, Song L, Crawford G, Giresi P, Lieb J, Liu Z, McDaniell R, Lee B, Iyer V, Flicek P, Keefe D, Birney E, Graf S | 2011 | Open Chromatin by DNaseI HS from ENCODE/OpenChrom(Duke University) | https://www.ncbi.nlm.nih.gov/geo/query/acc.cgi?acc=GSE32970 | Publicly available at the NCBI Gene Expression Omnibus (accession no: GSE32970) |
| Rouskin S, Zubradt M, Washietl S, Kellis M, Weissman JS | 2013 | Genome-wide probing of RNA structure reveals active unfolding of mRNA structures in vivo | https://www.ncbi.nlm.nih.gov/geo/query/acc.cgi?acc=GSE45803 | Publicly available at the NCBI Gene Expression Omnibus (accession no: GSE45803) |
| Sandstrom R | 2011 | DNaseI Hypersensitivity by Digital DNaseI from ENCODE/University of Washington | https://www.ncbi.nlm.nih.gov/geo/query/acc.cgi?acc=GSE29692 | Publicly available at the NCBI Gene Expression Omnibus (accession no: GSE29692) |
| Snyder M, Gerstein M, Weissman S, Farnham P, Struhl K | 2011 | ENCODE Transcription Factor Binding Sites by ChIP-seq from Stanford/Yale/USC/Harvard | https://www.ncbi.nlm.nih.gov/geo/query/acc.cgi?acc=GSE31477 | Publicly available at the NCBI Gene Expression Omnibus (accession no: GSE31477) |
| Descostes N, Heidemann M, Spinelli L, Schüller R, Maqbool MA, Fenouil R, Koch F, Innocenti C, Gut M, Gut I, Eick D, Andrau J | 2014 | Tyrosine phosphorylation of RNA Polymerase II CTD is associated with antisense promoter transcription and active enhancers in mammalian cells | https://www.ncbi.nlm.nih.gov/geo/query/acc.cgi?acc=GSE52914 | Publicly available at the NCBI Gene Expression Omnibus (accession no: GSE52914) |
| Liu W, Ma Q | 2014 | Brd4 and JMJD6-associated Anti-pause Enhancers in Regulation of Transcriptional Pause Release | https://www.ncbi.nlm.nih.gov/geo/query/acc.cgi?acc=GSE51633 | Publicly available at the NCBI Gene Expression Omnibus (accession no: GSE51633) |
| Chen F, Woodfin AR, Shilatifard A | 2015 | PAF1, a molecular regulator of promoter-proximal pausing by RNA Polymerase II | https://www.ncbi.nlm.nih.gov/geo/query/acc.cgi?acc=GSE70408 | Publicly available at the NCBI Gene Expression Omnibus (accession no: GSE70408) |

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
