## [Decision Letter]

Thank you for submitting your article "CDK9-dependent RNA polymerase II pausing controls transcription initiation" for consideration by *eLife*. Your article has been reviewed by three peer reviewers, one of whom is a member of our Board of Reviewing Editors and the evaluation has been overseen by Kevin Struhl as the Senior Editor. The reviewers have opted to remain anonymous.

The reviewers have discussed the reviews with one another and the Reviewing Editor has drafted this decision to help you prepare a revised submission.

In this manuscript, 4sU-seq and NET-seq were used in combination with an analogue-sensitive CDK9 mutant to investigate the relationship between transcriptional pausing and initiation. The reviewers agree that the findings definitively show that transcription initiation rates vary at strongly versus weakly paused RNA polymerase II promoters, and that inactivation of CDK9 thus affects transcription initiation, whether directly or indirectly, in a manner that correlates with pausing strength. The reviewers consider these conclusions extremely important and timely, and help explain how strongly paused genes can effectively and synchronously upregulate mRNA levels.

Comments are provided for revision:

1) Prior cell-free transcription experiments have shown large effects on mRNA synthesis due to pause-release under conditions that do not support enhancer-linked transcription initiation or chromatin templates to facilitate transcription reinitiation. The differences in pause frequency and strength between different promoters can also be recapitulated in vitro in the absence of long-range connections between enhancers or chromatin- the resulting changes in mRNA levels in these studies are quite large and P-TEFb/NELF-DSIF-dependent. Should the changes in initiation secondary to pausing observed here be more accurately described as indirect? Or would 4sU-seq and NET-seq reveal a stronger effect on initiation in these systems?

2) The authors should comment on the fact that the estimates for pause duration and initiation frequency in subsection “Multi-omics analysis provides pause duration *d* and initiation frequency *I*” disregard any differences there might be in transcriptional processivity between different genes and regions.

3) Why are the position-markers truncated in 'Figure 1' and 'Figure 2—figure supplement 1'? The rulers should cover the whole genomic area shown in the figure above.

4) Introduction: Consider inserting 'underlying' so the sentence reads: "The mechanisms underlying how Pol II pausing can regulate….."

5) Cite Saponaro et al., 2014, who also published on elongation speeds using the techniques referenced in subsection “Pol II elongation velocity is gene-specific”.

6) The authors have not directly analyzed initiation or the formation of pre-initiation complexes as was done by Shao and Zeitlinger, 2017. As such, they should use the term "pause release" rather than 'initiation frequency' given that the data are focused on pol II release from the pausing site. 7)

7) In subsection “TT-seq monitors immediate response to CDK9 inhibition”, they can't claim that pol II that is still going through the EEC after CDK9 inhibition is independent of CDK9 function. Is it not more likely that not every CDK9 has been fully inhibited by 10 min treatment of NA? The only (indirect) experimental evidence they have showing that pol II pausing is regulating pol II initiation at the promoter is the decrease in TT-seq signal between the TSS and the pause site after CDK9 inhibition. The problem here is that it is not entirely clear whether CDK9 activity is doing something upstream of the pausing site and is involved in elongation between the TSS and the pause site.

---

## [Author Response]

Comments are provided for revision:1) Prior cell-free transcription experiments have shown large effects on mRNA synthesis due to pause-release under conditions that do not support enhancer-linked transcription initiation or chromatin templates to facilitate transcription reinitiation. The differences in pause frequency and strength between different promoters can also be recapitulated in vitro in the absence of long-range connections between enhancers or chromatin- the resulting changes in mRNA levels in these studies are quite large and P-TEFb/NELF-DSIF-dependent. Should the changes in initiation secondary to pausing observed here be more accurately described as indirect? Or would 4sU-seq and NET-seq reveal a stronger effect on initiation in these systems?

We appreciate these insightful comments. We carefully considered whether we should describe the changes as indirect. We wish to maintain the current wording. One may argue that steric exclusion of PIC assembly is a direct, not an indirect, effect. Also, it may be that the word indirect leads readers to think that one must assume additional mechanisms, such as posttranslational modification etc. to explain the effect of pausing on initiation, but this is not the case.

2) The authors should comment on the fact that the estimates for pause duration and initiation frequency in subsection “Multi-omics analysis provides pause duration d and initiation frequency I” disregard any differences there might be in transcriptional processivity between different genes and regions.

This is an important point. It is correct that transcriptional processivity is not modeled in a gene-wise manner as far as if it happens at the pause site. However, early termination that might occur upstream of the pause site is irrelevant to our model and its conclusions. Additionally, Figure 4—figure supplement 1 shows that early termination events at the pause site occur rarely if at all. The minor role that transcriptional processivity might play is also negligible for the conclusions we draw on metagene analysis, involving 500 up to more than 2000 genes. We nevertheless made sure that we point out in the main text that the possibility exists that a fraction of paused polymerases is lost, i.e. processivity at this step is not 100%.

3) Why are the position-markers truncated in 'Figure 1' and 'Figure 2—figure supplement 1'? The rulers should cover the whole genomic area shown in the figure above.

We thank the reviewers for this helpful comment. We changed the figures accordingly.

4) Introduction: Consider inserting 'underlying' so the sentence reads: "The mechanisms underlying how Pol II pausing can regulate….."

We have added this change to the main text.

5) Cite Saponaro et al., 2014, who also published on elongation speeds using the techniques referenced in subsection “Pol II elongation velocity is gene-specific”.

We thank the reviewers for pointing this out. We have added this citation.

6) The authors have not directly analyzed initiation or the formation of pre-initiation complexes as was done by Shao and Zeitlinger, 2017. As such, they should use the term "pause release" rather than 'initiation frequency' given that the data are focused on pol II release from the pausing site. 7)

Shao and Zeitlinger exclusively used occupancy measurements of transcription factors. Note that an increase in occupancy can stem from a higher initiation frequency, but also from a longer live time of the PIC or from filling of unbound promoters in a cell population. TT-seq however measures RNA production at each genomic location. The TT-seq signal in genes measures the number of RNA-producing polymerases that initiated from the promoter and were successfully released form the pause site, i.e. the ‘productive initiation frequency’. Using the term ‘pause release’ for the initiation frequency would lead to many misunderstandings. The term ‘pause release rate’ would refer to the rate at which a paused polymerase is released, but this is conceptually different from the productive initiation frequency, and it is a critical point of the paper to say this. In order to avoid such confusion, we wish to maintain the term ‘productive initiation frequency’. It is true that this is only identical to the initiation frequency if no paused polymerases undergo early termination, but we show that the fraction of such early terminating polymerases is low. To account for the possibility of a fraction of Pol II terminating early, we added the term ‘productive’ to the term ‘initiation frequency’. We went through the entire text again and made sure it is all consistent, the right terminology is used throughout, and that misunderstandings are avoided.

7) In subsection “TT-seq monitors immediate response to CDK9 inhibition”, they can't claim that pol II that is still going through the EEC after CDK9 inhibition is independent of CDK9 function. Is it not more likely that not every CDK9 has been fully inhibited by 10 min treatment of NA? The only (indirect) experimental evidence they have showing that pol II pausing is regulating pol II initiation at the promoter is the decrease in TT-seq signal between the TSS and the pause site after CDK9 inhibition. The problem here is that it is not entirely clear whether CDK9 activity is doing something upstream of the pausing site and is involved in elongation between the TSS and the pause site.

The reviewer is correct that we cannot be certain of this point. We have therefore included a corresponding statement that we cannot exclude that CDK9 inhibition was incomplete. However, according to the distribution shown in Figure 1, and the additional genome browser views pasted below (see Author response image 1), where the response is around 90%, the portion of CDK9 that has not been fully inhibited by 10 min treatment with 1-NA-PP1 must be very low. This conclusion is based on the assumption that the inhibitor is evenly distributed across cells and within cells. Note also that inhibitory effects on pausing observed until now might have been also caused by unspecific inhibition of other kinases.

**Author response image 1. respfig1:** Example genome browser views of TT-seq signals in CDK9^as^ cells with high responsiveness (~ 90%). (**A**) CYB5R4 gene locus (107,781 [bp]) on chromosome 6. The upper panel shows TT-seq signal with CDK9^as^ inhibitor (red) and control (black). Grey box depicts transcript body from transcription start site (TSS, black arrow) to polyA site (pA). (**B**) AGPAT6 gene locus (47,814 [bp]) on chromosome 8 depicted as in (**A**). (**C**) PYGB gene locus (49,945 [bp]) on chromosome 20 depicted as in (**A**).